# Application of mini-MLST and whole genome sequencing in low diversity hospital extended-spectrum beta-lactamase producing *Klebsiella pneumoniae* population

**Matej Bezdicek**[1,2], **Marketa Nykrynova**[1,3], **Kristina Plevova**[1], **Eva Brhelova**[1,2], **Iva Kocmanova**[4], **Karel Sedlar**[3], **Zdenek Racil**[1,2], **Jiri Mayer**[1,2], **Martina Lengerova**[1,2]*

**1** Department of Internal Medicine–Hematology and Oncology, University Hospital Brno, Brno, Czech Republic, **2** Department of Internal Medicine–Hematology and Oncology, Masaryk University, Brno, Czech Republic, **3** Department of Biomedical Engineering, Brno University of Technology, Brno, Czech Republic, **4** Department of Clinical Microbiology, University Hospital Brno, Brno, Czech Republic

* Lengerova.Martina@fnbrno.cz

**Data Availability Statement:** Data accession for our project was updated in the manuscript. All raw

## Abstract

Studying bacterial population diversity is important to understand healthcare associated infections' epidemiology and has a significant impact on dealing with multidrug resistant bacterial outbreaks. We characterised the extended-spectrum beta-lactamase producing *K. pneumoniae* (ESBLp KPN) population in our hospital using mini-MLST. Then we used whole genome sequencing (WGS) to compare selected isolates belonging to the most prevalent melting types (MelTs) and the colonization/infection pair isolates collected from one patient to study the ESBLp KPN population's genetic diversity. A total of 922 ESBLp KPN isolates collected between 7/2016 and 5/2018 were divided into 38 MelTs using mini-MLST with only 6 MelTs forming 82.8% of all isolates. For WGS, 14 isolates from the most prominent MelTs collected in the monitored period and 10 isolates belonging to the same MelTs collected in our hospital in 2014 were randomly selected. Resistome, virulome and ST were MelT specific and stable over time. A maximum of 23 SNV per core genome and 58 SNV per core and accessory genome were found. To determine the SNV relatedness cut-off values, 22 isolates representing colonization/infection pair samples obtained from 11 different patients were analysed by WGS with a maximum of 22 SNV in the core genome and 40 SNV in the core and accessory genome within pairs. The mini-MLST showed its potential for real-time epidemiology in clinical practice. However, for outbreak evaluation in a low diversity bacterial population, mini-MLST should be combined with more sensitive methods like WGS. Our findings showed there were only minimal differences within the core and accessory genome in the low diversity hospital population and gene based SNV analysis does not have enough discriminatory power to differentiate isolate relatedness. Thus, intergenic regions and mobile elements should be incorporated into the analysis scheme to increase discriminatory power.

sequencing data are available under the accession number PRJNA515630 in the BioProject database (direct link was also added to the revised manuscript https://www.ncbi.nlm.nih.gov/bioproject/PRJNA515630.).

**Funding:** This study was supported by the Ministry of Health, Czech Republic - conceptual development of research organization (FNBr, 65269705)(MB, ML, IK, ZR, JM, KP, EB), Masaryk University Grant Agency (MUNI/A/1105/2018)(MB) and Czech Science Foundation (GACR 17-01821S) (MB, ML, MN, KS, IK). The funders had no role in study design, data collection and analysis, decision to publish, or preparation of the manuscript.

**Competing interests:** The authors have declared that no competing interests exist.

# Introduction

*Klebsiella pneumoniae* (KPN) frequently causes community and hospital acquired infections including pneumonia, urinary tract infections and pyogenic liver abscesses [1, 2]. The main KPN transmission reservoirs are the gastrointestinal tract and the hands of hospital personnel and patients. Nosocomial KPN isolates often display highly resistant phenotypes with an extended-spectrum beta-lactamases producing (ESBLp) KPN prevalence between 2% and 55% [3–5].

Typing methods to discriminate different bacterial isolates from the same species are essential epidemiological tools. In populations with a high prevalence of ESBL, the knowledge of bacterial population structure and dynamics is especially important in outbreak detection and intervention. To monitor the bacterial population, cheap, rapid and robust methods are needed. In our previous study, we proved that mini-MLST, a method derived from multi locus sequence typing (MLST) in which costly and time-consuming sequencing is replaced with high resolution melting analysis, is suitable for long term prospective KPN population screening. Currently, besides KPN [6], mini-MLST has been established for *Staphyloccoccus aureus* [7], *Enterococcus faecium* [8] and *Streptococcus pyogenes* [9]. However, its lower discriminatory power makes mini-MLST insufficient to identify outbreak strains and it needs to be combined with more sensitive methods.

Recently, whole genome sequencing (WGS) has revolutionized our ability to differentiate between bacterial strains at the entire genome's DNA sequence level. For bacterial typing, WGS has two major approaches–core genome multilocus sequence typing (cgMLST) and single nucleotide variant analysis (SNV) [10]. CgMLST is based on an allele numbering system of a pre-determined set of genes. An advantage of cgMLST's approach is its inter-laboratory portability and the existence of public databases e.g. https://www.cgmlst.org/ncs, http://enterobase.warwick.ac.uk/ or https://pubmlst.org/. SNV analysis is based on mapping raw sequence reads against a reference genome and detecting nucleotides that vary within the dataset. This approach provides an even higher resolution power than cgMLST, but is far more computationally intensive than cgMLST analysis and interpreting the results is more complex [11, 12].

As generating WGS data become more accessible, rapid and cheap, bottlenecks remain in proper pre-sequencing sample selection and post-sequencing data analysis [11]. The main bottlenecks include the need to critically evaluate the raw sequencing data, some knowledge and skills in programming and improvements in data analysis to translate the enormous amount of obtained data into understandable results for health professionals [13]. Knowledge of the local bacterial population's genetics characterisation is also crucial for the results to be correctly interpreted, as there are no general thresholds of relatedness [10].

The main objectives of this study were i) to characterise the ESBLp KPN population in our hospital using mini-MLST prospective typing ii) to evaluate core and accessory genome single nucleotide variant analysis contribution in possible outbreak detection within the low diversity ESBLp KPN hospital population.

# Material and methods

## Clinical isolates

The study was conducted at the University Hospital Brno (Brno, Czech Republic), a tertiary care hospital with more than 2,000 beds and 5,000 employees. There are more than 1,000,000 people treated in out-patient clinics and over 70,000 patients hospitalized every year. During the systematic strain collection between 7/2016 and 5/2018, we collected all ESBLp KPN

isolates from high risk departments (Department of Internal Medicine–Hematology and Oncology, Department of Internal Medicine, Geriatrics and Practical Medicine, Department of Anesthesiology and Intensive Care Medicine) and all isolates from neonates, new-borns and children. In total, we collected 922 ESBLp KPN isolates (Table 1). For the purposes of this study, 24 ESBLp KPN isolates from our previous study collected between 1/2014 and 10/2014 were also included [14]. All isolates were collected during routine practice and were made completely anonymous.

## DNA isolation

For mini-MLST, bacterial genomic DNA (gDNA) was isolated using Chelex 100 Resin (Bio-Rad, USA). The Bacterial culture was homogenised in 100 µ of 5% w/v Chelex 100 Resin with a vortex. The suspension was incubated for 10 min at 100˚C and then centrifuged for 2 min at

**Table 1. Epidemiological data; n = 922.**

| Gender | no. of isolates (%) |
|---|---|
| Male | 510 (55.3) |
| Female | 412 (44.7) |
| **Patient age** | **mean (range)** |
| | 56 years (0 days-99 years) |
| **Hospitalisation/outpatient** | **no. of isolates (%)** |
| Hospitalisation | 863 (93.6) |
| Outpatient | 59 (6.4) |
| | **mean (range)** |
| Hospitalisation length | 27 days (0–724 days) |
| Hospitalisation length to collection | 11 days (0–369 days) |
| **Source** | **no. of isolates (%)** |
| Urine | 269 (29.2) |
| Rectum | 226 (24.5) |
| Blood culture | 96 (10.4) |
| Oral cavity swab | 84 (9.1) |
| Throat swab | 49 (5.3) |
| Sputum | 43 (4.7) |
| Perianal swab | 39 (4.2) |
| Wound swab | 30 (3.3) |
| Urinary catheter | 24 (2.6) |
| Skin swab | 13 (1.4) |
| Handprint | 11 (1.2) |
| Venous catheter | 10 (1.1) |
| Nose swab | 7 (0.8) |
| Stool | 6 (0.7) |
| Bronchoalveolar lavage fluid | 4 (0.4) |
| Vagina | 3 (0.3) |
| Ascites | 2 (0.2) |
| Suction catheter | 2 (0.2) |
| Ear swab | 1 (0.1) |
| Eye swab | 1 (0.1) |
| Stoma swab | 1 (0.1) |
| Urethra swab | 1 (0.1) |

15,500 rcf. A supernatant containing gDNA was transferred into a clean microtube. For WGS, gDNA was purified using DNeasy Blood & Tissue Kit (Qiagen GmbH). The DNA concentration was measured using NanoDrop (Thermo Scientific, USA).

### Mini-MLST

Mini-MLST was performed with primers described by Andersson, Tong [6] on a RotorGene 6000 platform (Corbett Research, Australia). The 20 μL reaction volume contained 10 μL 2× SensiFAST HRM mix (Bioline Reagents, UK), 0.4 μM of each primer, 1 μL of extracted genomic DNA (30 ng) and deionized water to a final volume of 20 μL. Thermo cycling parameters were: 95˚C for 3 min, 40 cycles of 95˚C for 5 s, 65˚C for 10 s and 72˚C for 20 s, then one cycle of 95˚C for 2 min and 50˚C for 20 s, followed by HRM ramping from 70 to 95˚C, increasing by 0.2˚C at each step. The results were interpreted using the current version of our conversion key, which is available for free download at http://www.cmbgt.cz/mini-mlst/t6353.

### Whole genome sequencing and analysis

Purified gDNA was shredded using S220 Focused-ultrasonicator (Covaris, USA). WGS libraries were prepared with KAPA HyperPrep Kits (Roche, Switzerland) and a quality check was performed using 2100 Bioanalyzer (Agilent Technologies, USA). The Illumina MiSeq platform was used for WGS and 250-bp paired-end sequencing was performed. The raw sequencing data were deposited in the NCBI BioProject database under the project ID PRJNA515630 (https://www.ncbi.nlm.nih.gov/bioproject/PRJNA515630). The obtained reads were quality checked using FastQC (Babraham Bioinformatics, UK) and assembled using Burrows-Wheeler Aligner [15]. Ridom SeqSphere+ (Ridom, DE) seed genome *Klebsiella pneumoniae* subsp. *pneumoniae* NTUH-K2044 (NC_012731.1) was used as the reference genome. To remove unmapped reads, reads with poor quality and duplicates, SAMtools were used [16]. After reference mapping, all positions with less than 10× coverage and all ambiguous positions (less common base represented at least 10% of bases in the target position) were removed from further analysis. UGENE software was used to obtain consensus sequences [17]. Resistome and virulome analysis was carried out using public online databases (http://www.genomicepidemiology.org/, http://bigsdb.pasteur.fr/). The cgMLST was performed using Ridom SeqSphere+ (Ridom, DE) with an incorporated KPN cgMLST scheme. This included 2,358 target genes whose alleles were used to generate a cgMLST dendogram. For the core genome SNV minimum spanning tree 2,042,000 aligned nucleotide sites were analysed. In total, 4,179,387 aligned nucleotide sites were accounted for by the core and accessory genome SNV minimum spanning tree analysis. The WGS data were used to determine the MLST type of all sequenced strains.

## Results

### Mini-MLST

For hospital population characterisation, we used mini-MLST to type all collected isolates. Overall, 38 different MelTs were identified among 922 ESBLp KPN isolates collected between 7/2016 and 5/2018. MelT145 (30.8%, n = 284), MelT26 (25.9%, n = 239), MelT132 (8.0%, n = 74), MelT139 (7.7%, n = 71), MelT269 (5.4%, n = 50), MelT281 (5.0%, n = 46), MelT266 (2.6%, n = 24) and MelT20 (2.3%, n = 21) were the most predominant (Fig 1). The remaining 30 MelTs were present in less than 2% of all isolates. Fig 1 shows the proportions of certain MelTs in our hospital population are quite stable (MelT145, MelT26), while an increase (MelT132) or decrease (MelT139) can be observed in others. Mini-MLST's discriminatory

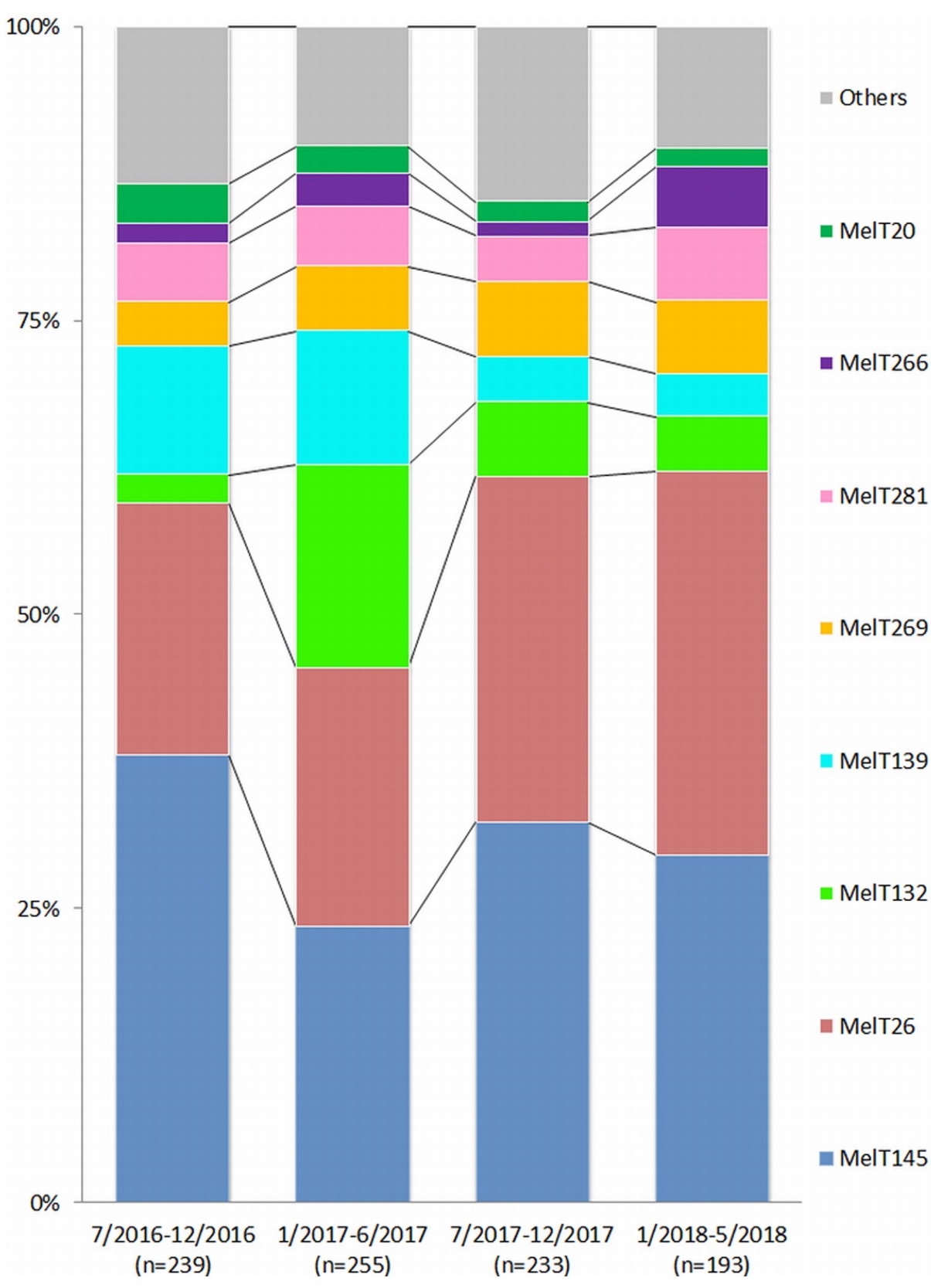

**Fig 1. Comparison of MelTs distribution in monitored period (7/2016-5/2018).**

power was D = 0.8189. The mini-MLST results were used for basic population sorting and WGS sample selection.

## Whole genome sequence analysis

**Whole genome sequence sample selection.**   We randomly selected 14 isolates from the four most predominant MelTs from 7/2016-5/2018 (3 MelT145 isolates, 5 MelT26 isolates, 3 MelT132 isolates and 3 MelT139 isolates). Because these MelTs were also present in our hospital during our previous study [14], we added 10 isolates isolated in our hospital in 2014 (3 MelT145 isolates, 1 MelT26 isolate, 3 MelT132 isolates and 3 MelT139 isolates).

To set SNV relatedness cut-off values, 22 isolates representing colonization (rectal swab)/ infection (blood culture) pair samples obtained from 11 different patients were analysed using WGS. Pair sample isolates belonged to 6 different MelTs (2 MelT26 isolates, 2 MelT130 isolates, 6 MelT132 isolates, 6 MelT145 isolates, 4 MelT266 isolates and 2 MelT281 isolates). For all 46 sequenced isolates, we performed *in silico* MLST, resistome and virulome analysis, cgMLST and SNV analysis of the core genome and accessory genome.

**The MLST, resistome and virulome analysis.**   First, *in silico* STs were determined and resistome and virulome were compared for all 46 samples (Fig 2). MLST analysis (based on seven house-keeping genes *rpoB*, *gapA*, *mhd*, *pgi*, *phoE*, *infB* and *tonB*) showed ST uniformity within individual MelT groups, except isolate S13 from MelT26 which, unlike the other isolates in this MelT group, was ST29. This isolate was the only MelT26 detected in 2014. All other sequenced MelT26 isolates from 2016 and 2017 were ST1271. ST1271 and ST29 only differ in the MLST scheme's *phoE* allele (allele *phoE* 4 in ST1271 against allele *phoE* 6 in ST26) (S1 Table).

Comparing resistome and virulome correlated with the MLST analysis results and confirmed a high level of similarity inside the MelTs (see detailed results in S1 Table). Isolates clustered together, as in the *in silico* MLST analysis (including separation of isolate S13 from other isolates) with one exception in MelT132, which was divided into two lineages, A and B, according to the different resistome and virulome composition. Lineage A included S19, S20, S21 and S44; lineage B included S22, S24, S25, S26, S35 and S36.

**cgMLST and SNV analysis.**   CgMLST analysis provides more discriminatory power than traditional MLST. In this study, we used a cgMLST scheme consisting of 2,358 targets processed by Ridom SeqSphere+ software (Ridom, DE). A cgMLST dendrogram based on 2,251 targets present in all analysed strains was constructed (Fig 2). The other 107 targets from the original 2,358 targets were absent in some or all of the strains and were excluded from all further analyses. The cgMLST divided 46 isolates into 7 clearly distinguishable clusters corresponding to their MelTs groups, with two exceptions. The S13 isolate differed from the other MelT26 isolates in 584 targets, while the average number of different alleles within the other MelTs ranged from 0 to 19 differences between two samples of the same MelT. The second exception was MelT132, which was divided into two subpopulations with 127 different alleles between them. Both exceptions correlated with the previous *in silico* MLST and resistome and virulome analysis. The average number of different alleles between individual MelTs was 1827 (range from 1809 to 1844 differences).

SNV analysis was carried out for both a core genome (n = 2,251 genes present in all analysed strains) and core with accessory genome (n = 4,084 genes present in all analysed strains). In the core genome, 29,535 SNV positions were identified in total. In general, the core genome SNV number inside each MelT varies from 0 to 23 SNV (MelT145 0–23 SNV, n = 12; MelT139 0–21 SNV, n = 8; MelT26 0–20 SNV, n = 8; MelT266 0–14 SNV, n = 4; MelT132 cluster A 0–22 SNV, n = 5; MelT132 cluster B 0–4 SNV, n = 6; MelT281 0 SNV, n = 2 and MelT130 0

| cgMLST | Sample ID | Patient | Collection Date | Time distance (days) | Ward | Isolation source | Mini-MLST | MLST | Pair samples cg SNV | Pair samples accg and cg SNV |
|---|---|---|---|---|---|---|---|---|---|---|
| | S31 | 30 | 25.5.2014 | 90 | G | BC | MelT266 | ST323 | 0 | 2 |
| | S32 | 30 | 24.2.2014 | | F | RC | MelT266 | ST323 | | |
| | S27 | 31 | 25.5.2014 | 1 | D | BC | MelT266 | ST323 | 1 | 5 |
| | S28 | 31 | 24.5.2014 | | D | RC | MelT266 | ST323 | | |
| | S41 | 24 | 8.9.2016 | 4 | D | BC | MelT26 | ST1271 | 0 | 3 |
| | S43 | 24 | 5.9.2017 | | D | RC | MelT26 | ST1271 | | |
| | S15 | 25 | 23.4.2017 | - | B | BC | MelT26 | ST1271 | - | - |
| | S18 | 27 | 7.7.2017 | - | C | VC | MelT26 | ST1271 | - | - |
| | S14 | 28 | 11.7.2016 | - | A | BC | MelT26 | ST1271 | - | - |
| | S16 | 26 | 6.6.2017 | - | D | RC | MelT26 | ST29 | - | - |
| | S17 | 23 | 22.6.2017 | - | D | BC | MelT26 | S1271 | - | - |
| | S13 | 29 | 7.6.2014 | - | E | UR | MelT26 | ST1271 | - | - |
| | S7 | 18 | 11.2.2016 | - | K | BC | MelT145 | ST433 | - | - |
| | S9 | 19 | 23.9.2014 | - | D | UR | MelT145 | ST433 | - | - |
| | S42 | 15 | 22.9.2014 | 10 | D | RC | MelT145 | ST433 | 0 | 0 |
| | S11 | 15 | 1.10.2014 | | D | BC | MelT145 | ST433 | | |
| | S2 | 17 | 17.3.2014 | - | J | WO | MelT145 | ST433 | - | - |
| | S39 | 22 | 12.8.2017 | 47 | D | BC | MelT145 | ST433 | 11 | 40 |
| | S38 | 21 | 23.9.2017 | 4 | D | BC | MelT145 | ST433 | 2 | 3 |
| | S37 | 21 | 26.9.2017 | | D | RC | MelT145 | ST433 | | |
| | S40 | 22 | 27.6.2017 | 47 | D | RC | MelT145 | ST433 | 11 | 40 |
| | S5 | 16 | 19.9.2016 | - | D | RC | MelT145 | ST433 | - | - |
| | S12 | 20 | 5.10.2016 | - | D | OC | MelT145 | ST433 | - | - |
| | S45 | 1 | 18.8.2014 | 71 | F | BC | MelT130 | ST458 | 0 | 1 |
| | S46 | 1 | 9.6.2014 | | A | RC | MelT130 | ST458 | | |
| | S19 | 2 | 11.3.2014 | - | A | OC | MelT132 | ST405 | - | - |
| | S20 | 3 | 21.9.2014 | 17 | D | BC | MelT132 | ST405 | 0 | 1 |
| | S21 | 3 | 21.9.2014 | | D | UR | MelT132 | ST405 | | |
| | S44 | 3 | 7.10.2014 | | D | RC | MelT132 | ST405 | | |
| | S23 | 5 | 22.5.2017 | - | D | OC | MelT132 | ST405 | - | - |
| | S22 | 4 | 24.4.2017 | - | D | BC | MelT132 | ST405 | - | - |
| | S25 | 7 | 15.2.2017 | 8 | D | RC | MelT132 | ST405 | 0 | 1 |
| | S24 | 6 | 31.5.2017 | - | H | UR | MelT132 | ST405 | - | - |
| | S26 | 7 | 22.2.2017 | 8 | D | BC | MelT132 | ST405 | 0 | 1 |
| | S35 | 8 | 17.12.2017 | 78 | D | BC | MelT132 | ST405 | 3 | 3 |
| | S36 | 8 | 10.10.2017 | | D | RC | MelT132 | ST405 | | |
| | S1 | 9 | 16.1.2017 | - | H | UR | MelT139 | ST321 | - | - |
| | S3 | 10 | 15.9.2016 | - | D | RC | MelT139 | ST321 | - | - |
| | S8 | 13 | 4.10.2016 | - | I | WD | MelT139 | ST321 | - | - |
| | S4 | 11 | 26.9.2014 | - | D | RC | MelT139 | ST321 | - | - |
| | S6 | 12 | 20.5.2014 | - | F | BC | MelT139 | ST321 | - | - |
| | S10 | 14* | 9.10.2014 | | M | BC | MelT139 | ST321 | - | - |
| | S29 | 14* | 9.10.2014 | 6 | M | BC | MelT139 | ST321 | 1 | 4 |
| | S30 | 14* | 4.10.2014 | | L | UR | MelT139 | ST321 | | |
| | S34 | 32 | 11.8.2014 | 35 | D | RC | MelT281 | ST23 | 0 | 1 |
| | S33 | 32 | 14.9.2014 | | D | BC | MelT281 | ST23 | | |

0.1

**Fig 2. WGS results of 46 selected ESBLp KPN isolates.** The cgMLST dendogram was made using Ridom Seqsphere+ software and is based on sequence similarity in the 2,251 core genome targets shared by all 46 isolates. Mini-MLST was done experimentally and MLST was done in silico using WGS data. Mini-MLST, MLST, cgMLST and SNV analysis of pair samples (blood culture/rectal swab pairs are highlighted in the same colour). The time lapse indicates time between collecting the colonizing and infecting isolates. * Samples obtained from one patient, but with no rectal swab.

SNV, n = 2) (Fig 3A). The S13 isolate differed by 2,311 SNV from other MelT26 isolates, while between individual MelTs, the number of SNV ranged from 9,595 to 9,821. The MelT132 isolates were divided into two lineages, A and B, similar to the cgMLST analysis. Both MelT132 lineages differed with 137 SNV between them. The complete distance matrix for the core genome SNV analysis is showed in S2 Table.

The core and accessory genome SNV analysis was done with 4,084 gene targets present in all 48 isolates (Fig 3B). 61,343 SNV positions were identified with the SNV count within each MelT ranging from 0 to 58 (MelT145 0–58 SNV, n = 12; MelT139 0–53 SNV, n = 8; MelT26 3–32 SNV, n = 8; MelT266 2–50 SNV, n = 4; MelT132 lineage A 0–36 SNV, n = 5; MelT132 lineage B 1–6 SNV, n = 6; MelT281 1 SNV, n = 2 and MelT130 1 SNV, n = 2). The average SNV number between individual MelTs was 20,196 SNV (range from 19,936 to 20,358 SNV). The distance matrix for the core and accessory genome's SNV analysis is showed in S3 Table. The major topology difference between the core (A) and the core and accessory (B) genome panels was the position exchange between the MelT281 and the MelT132 clusters. Additionally, using the core and accessory genome panel (B), the MelT26 and the MelT132 clusters were linked with the S13 isolate that was only linked to MelT26 cluster when using the core genome panel (A).

**Infection and colonisation pair isolates analysis.** We analysed 11 pairs (each pair included a rectal swab and a blood culture) of ESBLp KPN isolates obtained from 11 patients to set the SNV cut-off, which determined the relatedness of isolates (Fig 2). From 22 ESBLp KPN isolates, 6 distinct STs were identified: ST405 (n = 6), ST433 (n = 6), ST323 (n = 4), ST1271 (n = 2), ST458 (n = 2) and ST23 (n = 2). The isolates from each patient share the same ST within the pair. The core genome's SNV number within each pair varies in range from 0 to 22 SNV, while together with the accessory genome's SNV number, varies in range from 0 to 40 SNV. Based on paired sample analysis, we established relatedness cut-off values for our ESBLp KPN population as the highest SNV number within pair isolates, 22 SNV for the core genome and 40 SNV for the core with the accessory genome. Both the highest SNV values were for pair S39/S40 that had time lapses between samples of 47 days. The next highest SNV values were only 3 SNVs for the core genome and 5 SNVs for the core and accessory genome, despite time lapses of up to 90 days."

## Discussion

We are currently using the following protocol in routine practice. ESBLp KPN collected from high-risk departments are prospectively tested with mini-MLST to determine the MelT. When the strains MelT differ, transmission is unlikely. When we observe an increased incidence of one MelT, we investigate the potential epidemiological linkages and then we decide if there is a possible outbreak and need for WGS analysis. Meanwhile, early epidemiological measures can be implemented to prevent further spread of infection.

KPN mini-MLST is a cheap, rapid and robust method for epidemiological strain typing introduced by Andersson, Tong [6], with advantages in its robustness, reproducibility and portability between laboratories as it is based on a well-established MLST method. In studies with a large number of samples, mini-MLST also can be used for sample sorting and pre-selecting for more detailed analysis. We evaluated this method for prospective ESBLp KPN

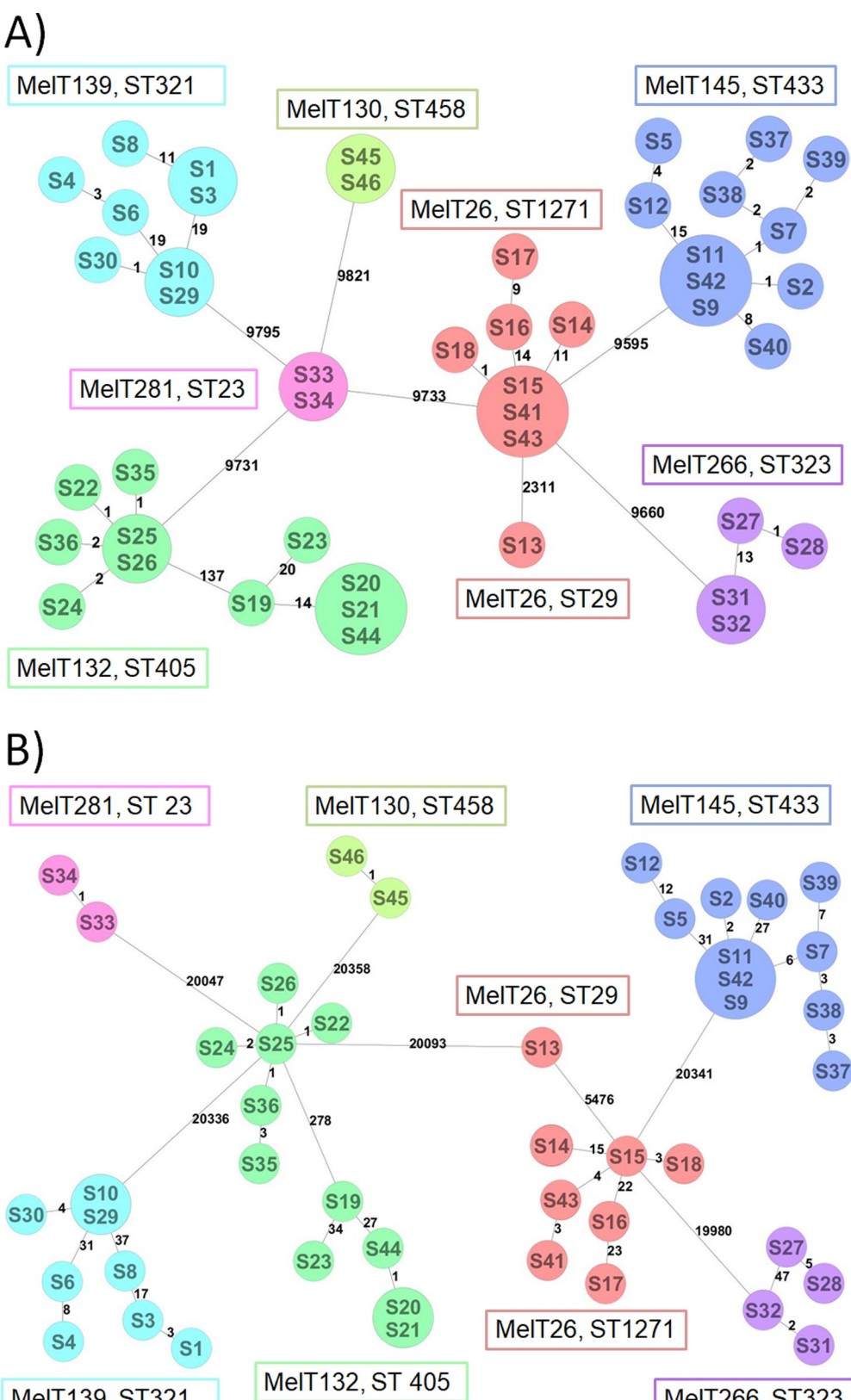

**Fig 3. Core genome SNV (A) and core with accessory genome SNV (B) analysis of 46 selected ESBLp KPN isolates.** SNV analysis was performed on sequences of 2,251 core genome targets and 4,084 core and accessory genome targets respectively. We found 29, 535 SNV positions in the core genome targets (from a total of 2,042,000 positions) and 61,343 SNV positions in the core and accessory targets (from a total of 4,179,387 positions).

population screening and to solve healthcare associated infection outbreaks in our previous study [14]. During the mini-MLST results evaluation, we were unable to determine the MelT of several isolates as their mini-MLST allele combination did not corresponded to any MelT in the published conversion key. On the date of the original article publication there were 863 STs, while in 7/2019 there were 4126 STs, from which it was not possible to determine some of the newly described STs' MelT according to the original conversion key. Following this, we constructed a new automatic algorithm (applicable to already existing and future-designed mini-MLST schemes) and created a new conversion key that will be regularly updated. The latest version of the key is available on http://www.cmbgt.cz/mini-mlst/t6353.

Based on our previous study, we started prospective screening in certain departments of our hospital in 7/2016. Since then, we have observed six MelTs account for 82.8% of isolates, from a total of 38 MelTs identified between 7/2016 and 5/2018 (Fig 1). The most pronounced changes in proportional representation were observed in MelT145 (from 38.1% in 2016 to 29.5% in 2018), MelT139 (from 10.9% in 2016 to 3.6% in 2018), MelT132 (from 2.5% in 2016 to 4.7% in 2018) and MelT266 (from 1.7% in 2016 to 5.2% in 2018). We did not detect any emerging MelT which spread to more than 1% of the study population. This suggests a relatively stable low diversity bacterial population which cannot be divided more by using the mini-MLST method only and needs methods with more discriminatory power.

WGS has become a powerful method for most epidemiological studies and hospital outbreak investigations, as it provides multiple analyses from a single technique, including data on MLST, resistant genes, virulence genes, plasmids and genome comparison [18]. Currently, epidemiological investigations have mainly focused on an allele-based approach (cgMLST, wgMLST) and SNV analysis. While the allele-based approach is particularly suitable for multi-centre studies and clustering large populations of bacteria [19], SNV analysis is especially useful when dealing with outbreak episodes where we are expecting the analysed strains to be similar. The number of differences in alleles or SNV that still identify isolates as similar or related and include them in outbreak episodes differs in various publications [11]. According to our best knowledge, there are no general standards or rules to set a sufficient threshold value. For this reason, it is not possible to simply compare the results of individual studies and freely use previously published cut-off values, the cut-off being typically determined based on the specific results from those publications and mostly varying between 0 and several dozen SNV [10]. Also, comparing outbreak strains and the local bacterial population is not commonly done in most studies, which may give important information to determine relatedness and assess outbreaks.

In order to evaluate the role of WGS in low diversity population outbreak analysis, we sequenced 46 ESBLp KPN isolates belonging to the four mostly spread MelTs from the monitored period from 7/2016 to 5/2018, isolates from the same MelTs originally collected for our mini-MLST pilot study in 2014 [14] and colonization/infection pair samples. First, we performed in silico MLST to correlate the STs with MelTs and characterise our population in the worldwide KPN population context. All predominant STs in our population have previously been described in literature. Serotype K1 strains belonging to ST23 (MelT281) were described as a frequent cause of invasive infections and liver abscesses [20]. ST321 (MelT139), ST323 (MelT266), ST1271 (MelT26) and ST29 (MelT26) were described as long-term, persistent MDR strains in hospital facilities [21–23]. ST405 (MelT132) was often described as an OXA-48

producer [24] and ST433 (MelT145) as hyper virulent biofilm producing strains [25]. The only discrepancy between MLST and mini-MLST was in sample 13, which belonged to ST29, in contrast with the other MelT26 isolates, which belonged to ST1271. S13 was the only MelT26 isolate collected in 2014.

Second, we used public online databases to determine the profile of resistance genes, virulence factor genes and efflux pump genes (S1 Table). Compared to the mini-MLST results, resistome, virulome and efflux pump gene analysis only separated S13 from others belonging to MelT26 (as well as MLST) and divided MelT132 into two clearly separate clusters (Isolates from clusters A and B from MelT132 both belonged to ST405). In summary, these analyses do not have sufficient discriminatory power for the epidemiology of low diversity hospital bacterial populations. However, they provide information on the resistance and virulence of the examined strains, which can help manage the spread of hyper-resistant and hyper-virulent strains and may be useful in patients' treatment.

The cgMLST provides more discriminatory power than previously mentioned analyses, since it is based on hundreds or thousands KPN genes' allelic similarities depending on the study's design [26–28]. Despite analysing 2,251 genes, we did not find enough allelic differences between isolates belonging to the same MelT to draw a conclusion about isolate relatedness.

To analyse the SNV analysis results, we have to set the SNV cut-off values to determine ESBLp KPN isolate similarity. We used the general premise that the colonization strain is in most cases the cause of the subsequent infection [26, 29, 30]. Therefore, the number of SNV between colonization/infection pair isolates defines the value of the difference level at which the isolates can still be considered similar. Based on our results, we established two sets of cut-off values. The first set was based on all pairs and cut-off values were 22 SNV for the core genome and 40 SNV for the core and accessory genome (Fig 2). The stricter cut-offs were set with excluded SNV values for pair S39/S40 and values were 3 SNV for the core genome and 5 SNV for the core and accessory genome. Both cut-off values set reflected the low diversity in the highly selective ESBLp KPN hospital population. Using both of our cut-off values on our population, even isolates without evident epidemiological associations were clustered together, which makes evaluating outbreaks difficult and may lead to erroneous conclusions.

We are aware that our study is a single-centre study with a small number of isolates sequenced. Especially to determine cut-off, more paired isolates should be analysed. However, due to the specific hospital's environment, we can expect highly-selected bacterial populations to also be found in other hospitals.

To our best knowledge, this is the first study where prospective molecular typing is combined with WGS to define the epidemiological background and the genetic structure of the hospital's bacterial population. We proved that mini-MLST is a cost effective means of ruling out epidemiological linkage, but only complete genome analysis can provide strong evidence in favour of epidemiological linkage. Our findings showed there were only minimal differences within the core/accessory genome in the low diversity hospital population and gene based SNV analysis does not have enough discriminatory power to differentiate isolates' relatedness or evaluate whether it is an outbreak or not. Thus, intergenic regions and mobile elements should be incorporated to the analysis scheme to increase the discriminatory power. Therefore, when evaluating any molecular biological data, it is necessary to analyse them to concord with the epidemiological background.

## Supporting information

**S1 Table. Resistome, virulome and efflux genes profiles of 46 ESBLp KPN isolates.**
(XLSX)

**S2 Table. Core genome SNV distance matrix.**
(XLSX)

**S3 Table. Core and accessory genome SNV distance matrix.**
(XLSX)

## Acknowledgments

We would like to thank to Petra Myskova and Pavel Roudnicky for collecting and storing the ESBLp KPN strains.

## Author Contributions

**Conceptualization:** Matej Bezdicek, Zdenek Racil, Martina Lengerova.

**Data curation:** Matej Bezdicek, Marketa Nykrynova, Eva Brhelova, Iva Kocmanova, Karel Sedlar, Zdenek Racil, Martina Lengerova.

**Formal analysis:** Martina Lengerova.

**Investigation:** Matej Bezdicek.

**Methodology:** Matej Bezdicek, Iva Kocmanova, Martina Lengerova.

**Project administration:** Matej Bezdicek, Martina Lengerova.

**Resources:** Kristina Plevova, Eva Brhelova, Iva Kocmanova.

**Software:** Marketa Nykrynova, Karel Sedlar.

**Supervision:** Zdenek Racil, Jiri Mayer, Martina Lengerova.

**Validation:** Matej Bezdicek, Kristina Plevova.

**Writing – original draft:** Matej Bezdicek, Martina Lengerova.

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
