## [Decision Letter · Decision Letter 0]

3 Jul 2019

PONE-D-19-14966

Application of whole genome sequencing in low diversity hospital extended-spectrum beta-lactamases producing Klebsiella pneumoniae population

PLOS ONE

Dear Dr. Lengerova,

Thank you for submitting your manuscript to PLOS ONE. After careful consideration, we feel that it has merit but does not fully meet PLOS ONE’s publication criteria as it currently stands. Therefore, we invite you to submit a revised version of the manuscript that carefully addresses each of the points raised during the review process.

We would appreciate receiving your revised manuscript by Aug 17 2019 11:59PM. To enhance the reproducibility of your results, we recommend that if applicable you deposit your laboratory protocols in protocols.io, where a protocol can be assigned its own identifier (DOI) such that it can be cited independently in the future. For instructions see: http://journals.plos.org/plosone/s/submission-guidelines#loc-laboratory-protocols

We look forward to receiving your revised manuscript.

Kind regards,

D. Ashley Robinson, Ph.D.

Academic Editor

PLOS ONE

**Journal Requirements:**

2. Our editorial staff has assessed your submission, and we have concerns about the grammar, usage, and overall readability of the manuscript.  We therefore request that you revise the text to fix the grammatical errors and improve the overall readability of the text before we send it for review. We suggest you have a fluent, preferably native, English-language speaker thoroughly copyedit your manuscript for language usage, spelling, and grammar.

If you do not know anyone who can do this, you may wish to consider employing a professional scientific editing service.  

Whilst you may use any professional scientific editing service of your choice, PLOS has partnered with both American Journal Experts (AJE) and Editage to provide discounted services to PLOS authors. Both organizations have experience helping authors meet PLOS guidelines and can provide language editing, translation, manuscript formatting, and figure formatting to ensure your manuscript meets our submission guidelines. To take advantage of our partnership with AJE, visit the AJE website (http://learn.aje.com/plos/) and enter referral code PLOS15 for a 15% discount off AJE services. To take advantage of our partnership with Editage, visit the Editage website (www.editage.com) and enter referral code PLOSEDIT for a 15% discount off Editage services. If the PLOS editorial team finds any language issues in text that either AJE or Editage has edited, the service provider will re-edit the text for free.

Please note that PLOS ONE does not copyedit accepted manuscripts and that one of our criteria for publication is that articles must be presented in an intelligible fashion and written in clear, correct, and unambiguous English (http://www.plosone.org/static/publication#language). If the language is not sufficiently improved, we may have no choice but to reject the manuscript without review.

**Comments to the Author**

1. Is the manuscript technically sound, and do the data support the conclusions?

Reviewer #1: Yes

Reviewer #2: Yes

2. Has the statistical analysis been performed appropriately and rigorously? 

Reviewer #1: Yes

Reviewer #2: N/A

3. Have the authors made all data underlying the findings in their manuscript fully available?

Reviewer #1: No

Reviewer #2: No

4. Is the manuscript presented in an intelligible fashion and written in standard English?

Reviewer #1: Yes

Reviewer #2: Yes

5. Review Comments to the Author

Reviewer #1: This manuscript is particularly well written and describes an interesting and well-designed study. My comments are quite minor.

• Not sure where the sequence data/sequence reads are deposited

• The figure legends are short and uninformative. I know they should not repeat was in the text, but I was having trouble understanding exactly how each figure was put together. Perhaps just a little more information is necessary. One thing that was specifically unclear was whether the dendrogram in Fig 3 was on the basis of allele number similarity, or overall sequence similarity. Similarly, it appears from the Methods that allele numbers were used to create the minimum spanning trees in Fig 4 – which seems odd given that the Results implies it is from actual sequence similarity. Can this be clarified?

• Can the authors give some thought to shortening the Discussion? The main points conclusions could possibly be stated more clearly and succinctly e.g. “miniMLST is a cost effective means of ruling out epidemiological linkage, but only complete genome analysis can provide strong evidence in favour of epidemiological linkage”.

• Line 61: should be “bacterial”

Reviewer #2: Title: Application of whole genome sequencing in low diversity hospital extended-spectrum beta-lactamases producing Klebsiella pneumoniae population

This manuscript presents a well written description of the application of mini-MLST in the surveillance of ESBL-KPN within a single hospital in the Czech Republic. Additionally, the authors discuss the limitations of mini-MLST in resolving closely related isolates, in particular outbreak strains, and the need for WGS in these situations. The authors then show the potential application of WGS in a limited sample set.

Minor comments:

Line 52 – the word “solution” should likely be “intervention”.

Line 237-238 – The authors indicate they have generated a new automatic algorithm that acts as a conversion key. Is it possible to utilize this conversion key with the existing MLST schema to generate a conversion list of MSLT to melT groups? This would allow for direct comparison of the large existing studies that use MLST with the use of melT groups and will also clarify the resolution of melT groups. I think this would help clarify the potential role of melT groups among a growing list of molecular epidemiology markers of bacterial diversity (e.g. PFGE, MLST, mini-MLST, cgMLST, WGS).

Line 63-65 – The authors define CgMLST and SNV analyses, but the descriptions are confusing. In particular, the authors indicate SNV analysis provides a higher resolution power as compared to cgMLST, but then use SNV as a precursor to CgMLST. It would be good to clarify that both are based upon the detection of variants, but while CgMLST is looking at variants in a specific pre-determined set of genes, true whole genome alignment allows for assessment of variants in all regions that are not masked.

Table 1.- Can the authors please include the percentage of each subgrouping with the count? This will help the reader understand the proportion of isolates, especially within the source section of the table.

Lines 95-97 – the methods listed for use with mini-MLST differ from the methods used in the previous reference within Diagnostic Microbiology and Infectious Disease (citation 14). Can the authors comment on why the methods were changed from the UltraClean Microbial DNA Isolation Kit to the Chelex based extraction?

Line 122 – Can the authors indicate how they defined ambiguous positions? Was this based upon a percent threshold for variant detection?

Line 235 – please update the number of ST types as it now appears there are 4,054 different MSLT profiles within the Pasteur MLST database.

Line 249 – Please update the sentence to say “WGS has become a powerful method”

Major comments:

Title – As this manuscript focuses heavily on mini-MLST, please add mini-MLST to the title.

Lines 35-37, 57-59, 318-323 – The authors describe the potential for mini-MLST in real-time epidemiology, but point out the need for mini-MLST to be combined with more discriminatory analyses in outbreak/low genetic diversity settings. Can the authors describe in more detail how this progressive application of mini-MLST and WGS would work in active surveillance? In specific, it would be good to identify how mini-MLST could be used to select isolates for WGS and how mini-MLST would be able to identify outbreaks to focus on.

Lines 146-156 – Do the authors intend to submit the WGS sequencing data to a repository such as the European Nucleotide Archive? If so, can the authors include the project ID within the manuscript?

Line 215-217 – What was the order of collection for the colonization and infection strains? Were colonization strains always collected before outbreak strains? This may be good information to include in a supplemental table. Additionally, can the authors discuss the number of remaining positions within the genome that are accounted for in calculating the number of SNVs between each pair of isolates?

Line 271-275 – Can the authors make it more clear if this discussion is referring to the 14 isolates described on lines 146-147, the 24 isolates described from lines 146-150, or does this refer to the entire set of 46 isolates (line 155)? If this is either the 14 or 24 isolates, this represents a small subset of the overall 922 isolates. Can the authors discuss the possibility that this is not a broad enough subsample of the total set of ESBL-KPN isolates to truly evaluate the role of WGS in a low diversity population?

Line 301-310 – I am confused about the application of the premise that “the colonization strain is the cause of the subsequent infection in most cases” to the idea of evaluating a hospital outbreak. In particular, hospital outbreaks are characterized by person-to-person transmission, or are mediated by fomites in the environment. Can the authors describe the potential for misclassification of colonizing and infecting strains if the infecting strain was acquired within the hospital environment?

In a time where there is more focus on acquired resistance genes, and in particular how these acquired genes can change the required antimicrobial treatment choices, how can a technique like mini-MLST be used to help target hospital infection control in the case of an outbreak?

Figure 2 – This figure is very data dense, but does not display well on a single printed page. Additionally, as the discussion of virulence factors is limited within the manuscript, it may be good to reformat this figure or move it to the supplemental files.

6. PLOS authors have the option to publish the peer review history of their article (what does this mean?). If published, this will include your full peer review and any attached files.

Reviewer #1: No

Reviewer #2: No

---

## [Author Response · Author response to Decision Letter 0]

15 Jul 2019

Journal Requirements 

• Language corrections 

Our manuscript was reviewed by a native English-speaker. His name is Matthew Smith at Brno Editing, Proofreading & Translation Services (www.translationandproofreadingservices.com) and his contact e-mail is matsmithj@yahoo.com.

Editor Comments to Author 

• Have the authors made all data underlying the findings in their manuscript fully available? 

Line 119-121 - Data accession for our project was updated in the manuscript. All raw sequencing data are available under the accession number PRJNA515630 in the BioProject database (direct link was also added to the revised manuscript https://www.ncbi.nlm.nih.gov/bioproject/PRJNA515630.) 

Reviewers' Comments to Author 

Reviewer: 1 

• Not sure where the sequence data/sequence reads are deposited 

Line 119-121 - Data accession for our project was updated in the manuscript. All raw sequencing data are available under the accession number PRJNA515630 in the BioProject database (direct link was also added to the revised manuscript https://www.ncbi.nlm.nih.gov/bioproject/PRJNA515630.) 

• The figure legends are short and uninformative. I know they should not repeat was in the text, but I was having trouble understanding exactly how each figure was put together. Perhaps just a little more information is necessary. One thing that was specifically unclear was whether the dendrogram in Fig 3 was on the basis of allele number similarity, or overall sequence similarity. Similarly, it appears from the Methods that allele numbers were used to create the minimum spanning trees in Fig 4 – which seems odd given that the Results implies it is from actual sequence similarity. Can this be clarified? 

Line 193-199; Line 213-217 - We added short legends to Fig 2 and Fig 3 (formerly Fig 3 and Fig 4). We also clarified allele/sequence similarity in the Methods and Results sections. We transferred Fig 2 to the Table and moved it to the supplemental files (S1 Table).

• Can the authors give some thought to shortening the Discussion? The main point’s conclusions could possibly be stated more clearly and succinctly e.g. “miniMLST is a cost effective means of ruling out epidemiological linkage, but only complete genome analysis can provide strong evidence in favour of epidemiological linkage”. 

Line 241-341 - We rewrote the discussion to highlight the main conclusions

• Line 61: should be “bacterial” 

Line 62 - Text corrected

Reviewer: 2 

Minor comments : 

• Line 52 – the word “solution” should likely be “intervention”. 

Line 53 - Text corrected

• Line 237-238 – The authors indicate they have generated a new automatic algorithm that acts as a conversion key. Is it possible to utilize this conversion key with the existing MLST schema to generate a conversion list of MSLT to melT groups? This would allow for direct comparison of the large existing studies that use MLST with the use of melT groups and will also clarify the resolution of melT groups. I think this would help clarify the potential role of melT groups among a growing list of molecular epidemiology markers of bacterial diversity (e.g. PFGE, MLST, mini-MLST, cgMLST, WGS). 

Line 259-260 - Once you have designed the mini-MLST scheme the algorithm is applicable to any MLST scheme. To run the algorithm, you need a list of ST including allele variants; FASTA file of all genes used in mini-MLST scheme and mini-MLST primers sequences. Added to the manuscript.

• Line 63-65 – The authors define CgMLST and SNV analyses, but the descriptions are confusing. In particular, the authors indicate SNV analysis provides a higher resolution power as compared to cgMLST, but then use SNV as a precursor to CgMLST. It would be good to clarify that both are based upon the detection of variants, but while CgMLST is looking at variants in a specific pre-determined set of genes, true whole genome alignment allows for assessment of variants in all regions that are not masked. 

Line 64-71 - We rewrote the definition of cgMLST and SNV analysis to clarify the differences between those two approaches.

• Table 1.- Can the authors please include the percentage of each subgrouping with the count? This will help the reader understand the proportion of isolates, especially within the source section of the table. 

Table 1, Line 96 - Subgroups percentages were added to the Table 1.

• Lines 95-97 – the methods listed for use with mini-MLST differ from the methods used in the previous reference within Diagnostic Microbiology and Infectious Disease (citation 14). Can the authors comment on why the methods were changed from the UltraClean Microbial DNA Isolation Kit to the Chelex based extraction? 

As we process a large number of isolates we were looking for a more efficient DNA extraction method. Chelex based extraction costs $ 0.5 per sample and takes less than 30 minutes with minimum hands-on time. UltraClean Microbial DNA Isolation Kit costs $ 5 per sample, takes about 60 minutes and requires more hands-on time. The quality and concentration of DNA extracted with Chelex is suitable for use in PCR methods. 

Line 122 – Can the authors indicate how they defined ambiguous positions? Was this based upon a percent threshold for variant detection? Line 127-128 - The ambiguous base was added to a position when there were at least two different bases and the less common base represented at least 10 % of bases in the target position. We put a note in the manuscript.

Line 235 – please update the number of ST types as it now appears there are 4,054 different MSLT profiles within the Pasteur MLST database. Line 257 - The number of MLST profiles was updated in manuscript on 9/7/2019.

Line 249 – Please update the sentence to say “WGS has become a powerful method” Line 271 - Text corrected

Major comments : 

Title – As this manuscript focuses heavily on mini-MLST, please add mini-MLST to the title. Title, line 1 - Title updated to “Application of mini-MLST and whole genome sequencing in low diversity hospital extended-spectrum beta-lactamases producing Klebsiella pneumoniae population”

Lines 35-37, 57-59, 318-323 – The authors describe the potential for mini-MLST in real-time epidemiology, but point out the need for mini-MLST to be combined with more discriminatory analyses in outbreak/low genetic diversity settings. Can the authors describe in more detail how this progressive application of mini-MLST and WGS would work in active surveillance? In specific, it would be good to identify how mini-MLST could be used to select isolates for WGS and how mini-MLST would be able to identify outbreaks to focus on. Line 242-247 - The following paragraph was added to the discussion.

We are currently using the following protocol in routine practice. ESBLp KPN collected from high-risk departments are prospectively tested with mini-MLST to determine MelT. When the strains’ MelT differ, transmission is unlikely. When we observe an increased incidence of one MelT, we check the potential epidemiological linkages and then we decide if there is a possible outbreak and need of WGS analysis. Meanwhile, early epidemiological measures can be implemented to prevent further spread of infection.

Lines 146-156 – Do the authors intend to submit the WGS sequencing data to a repository such as the European Nucleotide Archive? If so, can the authors include the project ID within the manuscript? Line 119-121 - Data accession for our project was updated in the manuscript. All raw sequencing data are available under the accession number PRJNA515630 in the BioProject database (direct link was also added to the revised manuscript https://www.ncbi.nlm.nih.gov/bioproject/PRJNA515630.) 

• Line 215-217 – What was the order of collection for the colonization and infection strains? Were colonization strains always collected before outbreak strains? This may be good information to include in a supplemental table. Additionally, can the authors discuss the number of remaining positions within the genome that are accounted for in calculating the number of SNVs between each pair of isolates? Fig 2, Line 193 - Collection dates are listed in Fig 2 and also in S1 Table. In one case the rectum isolate was collected three days before the blood culture isolate and in one case the rectum isolate was collected sixteen days before the blood culture isolate. All other colonization isolates were collected before blood culture isolates. 

In total 4,179,387 positions were used in SNV analysis. For core genome SNV analysis only, 2,042,000 positions were analysed. We added S2 and S3 Tables to supplementary files, which show SNV distance matrix for all 46 samples for core genome SNV and core and accessory genome SNV respectively.

• Line 271-275 – Can the authors make it more clear if this discussion is referring to the 14 isolates described on lines 146-147, the 24 isolates described from lines 146-150, or does this refer to the entire set of 46 isolates (line 155)? If this is either the 14 or 24 isolates, this represents a small subset of the overall 922 isolates. Can the authors discuss the possibility that this is not a broad enough subsample of the total set of ESBL-KPN isolates to truly evaluate the role of WGS in a low diversity population? 

Line 287-290 - This part of the discussion is referring to all 46 sequenced isolates. We clarified this in the relevant part of the discussion. 

Our further project will focus on more detailed analysis of the most predominant MelTs, including sequencing of a larger isolate set, long reads sequencing and also plasmid analysis.

• Line 301-310 – I am confused about the application of the premise that “the colonization strain is the cause of the subsequent infection in most cases” to the idea of evaluating a hospital outbreak. In particular, hospital outbreaks are characterized by person-to-person transmission, or are mediated by fomites in the environment. Can the authors describe the potential for misclassification of colonizing and infecting strains if the infecting strain was acquired within the hospital environment? 

Line 314-318 - The number of SNV between colonization/infection pair isolates defines the value of the difference level at which the isolates can still be considered. 

We clarified this in the relevant part of the discussion. 

In our study (and hospital), the infecting strain is collected from blood culture, urine or other primary sterile material. If the patient has an infection with his own colonizing strain, it is possible to tell only when we also have an isolate from this patient from a previous collection.

• In a time where there is more focus on acquired resistance genes, and in particular how these acquired genes can change the required antimicrobial treatment choices, how can a technique like mini-MLST be used to help target hospital infection control in the case of an outbreak? 

Using knowledge of the local population, its stability and epidemiological linkages along with rapid prospective screening can help implement initial epidemiological measures or find and eliminate a potential outbreak source. As such, the Mini-MLST does not provide information on antibiotic resistance genes. However, information on the resistant genes’ presence in certain MelT acquired by the WGS along with the fact that the hospital population seems to be relatively stable over the long term can be used to predict the resistome of the examined strains. However, to confirm, the strains resistome needs to be examined by specific PCR or WGS.

• Figure 2 – This figure is very data dense, but does not display well on a single printed page. Additionally, as the discussion of virulence factors is limited within the manuscript, it may be good to reformat this figure or move it to the supplemental files. 

S1 Table - We changed the format to Table and move it to the supplemental files

---

## [Editor Report · Decision Letter 1]

22 Jul 2019

PONE-D-19-14966R1

Application of mini-MLST and whole genome sequencing in low diversity hospital extended-spectrum beta-lactamases producing Klebsiella pneumoniae population

PLOS ONE

Dear Dr. Lengerova,

Thank you for submitting your manuscript to PLOS ONE. After careful consideration, we feel that it has merit but does not fully meet PLOS ONE’s publication criteria as it currently stands. The Academic Editor believes that the manuscript's remaining issues with English grammar and use can be addressed by the authors. Therefore, we invite you to submit a revised version of the manuscript that addresses the Editors comments below.

We would appreciate receiving your revised manuscript by Sep 05 2019 11:59PM. To enhance the reproducibility of your results, we recommend that if applicable you deposit your laboratory protocols in protocols.io, where a protocol can be assigned its own identifier (DOI) such that it can be cited independently in the future. For instructions see: http://journals.plos.org/plosone/s/submission-guidelines#loc-laboratory-protocols

We look forward to receiving your revised manuscript.

Kind regards,

D. Ashley Robinson, Ph.D.

Academic Editor

PLOS ONE

Additional Editor Comments (if provided):

Title

*should read "beta-lactamase producing", not plural "beta-lactamases producing"

Abstract

*lines 32, 35, 39, referring to the analysis of core and accessory genes as "core/accessory" is ambiguous in the abstract, please state as "core and accessory" as done elsewhere in the text

Intro

*first 2 sentences are very poorly constructed, please revise

*line 65, suggested wording "of a pre-determined set of.."

Methods

*lines 133, 134 suggested wording of "aligned nucleotide sites" instead of ambiguous term "positions"

Results

*where is Fig 1? I did not see it in the files attached to this paper

*line 175 previous wording is better, suggested wording "(including separation of isolate S13 from other isolates)"

*lines 187 and 190 "average number of different alleles" is more precise language, please revise as appropriate

*Fig3 shows a major topology difference between panels A (core) and B (core and accessory) in the switching of the position of the MelT281, ST23 isolates and MelT132, ST405 isolates. This should be mentioned.

*line 234 suggested wording "Based on paired sample analysis,..."

*lines 237 to 240 wording is unclear. Do the authors mean "Both the highest SNV values were for pair S39/S40 that had time lapses between samples of 47 days. The next highest SNV values were only 3 SNVs for the core and 5 SNVs for the core and accessory, despite time lapses of up to 90 days." Please revise as appropriate.

Discussion

*line 263 suggested wording "Since then, we have observed six MelTs that account for 82.8% of isolates, from a total of 38 MelTs..."

*line 276 instead of "big bacterial populations", suggested wording "large populations of bacteria"

Overall

*Fig2 legend, Same issue as raised by Reviewer1, is the tree based on number of shared alleles or the sequence similarity in the shared alleles? Mostly likely, this legend should include "is based on sharing of 2,251 core genome targets." Please revise as appropriate.

---

## [Author Response · Author response to Decision Letter 1]

30 Jul 2019

Title 

• Should read "beta-lactamase producing", not plural "beta-lactamases producing" 

o Line 1 - Title corrected

Abstract 

• Lines 32, 35, 39, referring to the analysis of core and accessory genes as "core/accessory" is ambiguous in the abstract, please state as "core and accessory" as done elsewhere in the text 

o Lines 32, 35 and 39 - Text corrected

Introduction 

• First 2 sentences are very poorly constructed, please revise 

o Line 44-49 – Text corrected

• Line 65, suggested wording "of a pre-determined set of.." 

o Line 65 – Text corrected

Methods 

• Lines 133, 134 suggested wording of "aligned nucleotide sites" instead of ambiguous term "positions" 

o Line 133 and 134 – Text corrected

Results 

• Where is Fig 1? I did not see it in the files attached to this paper 

o Fig 1 is uploaded in the attachment files

• Line 175 previous wording is better, suggested wording "(including separation of isolate S13 from other isolates)" 

o Line 175-176 – Text corrected

• Lines 187 and 190 "average number of different alleles" is more precise language, please revise as appropriate 

o Lines 187 and 190 – Text corrected

• Fig3 shows a major topology difference between panels A (core) and B (core and accessory) in the switching of the position of the MelT281, ST23 isolates and MelT132, ST405 isolates. This should be mentioned. 

o Lines 229-233 – This information was added to the manuscript

• Line 234 suggested wording "Based on paired sample analysis,..." 

o Line 239 – Text corrected

• Lines 237 to 240 wording is unclear. Do the authors mean "Both the highest SNV values were for pair S39/S40 that had time lapses between samples of 47 days. The next highest SNV values were only 3 SNVs for the core and 5 SNVs for the core and accessory, despite time lapses of up to 90 days." Please revise as appropriate. 

o Lines 242 to 244 – Text corrected

Discussion 

• Line 263 suggested wording "Since then, we have observed six MelTs that account for 82.8% of isolates, from a total of 38 MelTs..." 

o Line 267-268 – Text corrected

• Line 276 instead of "big bacterial populations", suggested wording "large populations of bacteria" 

o Line 280 – Text corrected

Overall 

• Fig2 legend, Same issue as raised by Reviewer1, is the tree based on number of shared alleles or the sequence similarity in the shared alleles? Mostly likely, this legend should include "is based on sharing of 2,251 core genome targets." Please revise as appropriate. 

o Line 194 – Fig 2 legend revised

---

## [Editor Report · Decision Letter 2]

1 Aug 2019

Application of mini-MLST and whole genome sequencing in low diversity hospital extended-spectrum beta-lactamases producing Klebsiella pneumoniae population

PONE-D-19-14966R2

Dear Dr. Lengerova,

We are pleased to inform you that your manuscript has been judged scientifically suitable for publication and will be formally accepted for publication once it complies with all outstanding technical requirements.

With kind regards,

D. Ashley Robinson, Ph.D.

Academic Editor

PLOS ONE
---

## [Editor Report · Acceptance letter]

5 Aug 2019

PONE-D-19-14966R2 

Application of mini-MLST and whole genome sequencing in low diversity hospital extended-spectrum beta-lactamase producing *Klebsiella pneumoniae* population 

Dear Dr. Lengerova:

I am pleased to inform you that your manuscript has been deemed suitable for publication in PLOS ONE. Congratulations! Your manuscript is now with our production department. 

With kind regards,

on behalf of

Dr. D. Ashley Robinson 

Academic Editor

PLOS ONE